# Effect of Gradient Energy Density on the Microstructure and Mechanical Properties of Ti6Al4V Fabricated by Selective Electron Beam Additive Manufacture

**DOI:** 10.3390/ma13071509

**Published:** 2020-03-26

**Authors:** Ta-I Hsu, Yu-Ting Jhong, Meng-Hsiu Tsai

**Affiliations:** 1The Genomics Research Center, Academia Sinica, Taipei 115, Taiwan; cardiosea@gmail.com; 2School of Dentistry, College of Dental Medicine, Kaohsiung Medical University, Kaohsiung 807, Taiwan; steelwillpower0130@gmail.com; 3Department of Mold and Die Engineering, National Kaohsiung University of Science and Technology, Kaohsiung 807, Taiwan

**Keywords:** selective electron beam additive manufacture, Ti6Al4V ELI alloy, phase transformation, spatial, gradient energy density, martensitic decomposition, Ti_3_Al intermetallic compound, fracture analysis

## Abstract

Selective Electron Beam Additive Manufacturing (SEBAM) is a promising powder bed fusion additive manufacturing technique for titanium alloys that select particular area melting in different energy density for producing complexly shaped biomedical devices. For most commercial Ti6Al4V porous medical devices, the gradient energy density is usually applied to manufacture in one component during the SEBAM process which selects different energy density built on particular zones. This paper presents gradient energy density base characterization study on an SEBAM built rectangular specimen with a size of 3 mm × 20 mm × 60 mm. The specimen was divided into three zones were built in gradient energy density from 16 to 26.5 J/mm^3^. The microstructure and mechanical properties were investigated by means of scanning electron microscopy, X-ray diffraction, transmission electron microscopy and mechanical test. The α′ martensitic and lack of fusion were observed in the low energy density (LED) built zone. However, no α′ phase and no irregular pores were observed both in overlap energy density (OED) and high energy density (HED) built zones located at the middle and bottom of the specimen respectively. This implies the top location and lower energy density have positive effects on the cooling rate but negative effects on densification. The subsequence mechanical properties result also supports this point. Moreover, the intermetallic Ti_3_Al found in the bottom may be due to the heat transfer from the following melting layer. Furthermore, the microstructure evolution in gradient energy built zones is discussed based on the findings of the microstructure and thermal history correlation analysis.

## 1. Introduction

Selective electron beam additive manufacture (SEBAM) is one of powder bed fusion technology that builds metallic parts with complex geometries by using the electron beam as the heat source to selectively preheat and melt over the metallic powder bed via layer by layer based on 3D CAD model input. SEBAM has attracted much attention over the past few years for its advantages, such as high material utilization, low porosity and fast production rate without post heat treatment in particular cases, and so on. Furthermore, SEBAM is an advanced manufacturing technology that has widely fabricated titanium, nickel base, CoCrMo and difficult forming alloys [1,2].

Ti6Al4V is commonly used in the SEBAM process. Its excellent corrosion resistance, mechanical property and biocompatibility are widely applied to medical applications such as orthopedic [3], dental [4] and spinal implants [5]. Porous medical metallic parts by SEBAM process, which possess low elastic Young’s modulus matching to bone tissue and are capable of providing space for in growth of bony tissue to achieve better fixation, have been widely used for medical implants due to reduced stress shielding effect [6] and increased bone osseointegration [7,8]. Therefore, most researchers adjusted the porosity and geometric to improve the benefit biocompatible reticulated meshes and foams with an interconnected porous structure which has porosity open cell and foams (porosity between 55%~90%) [9,10,11]. As mentioned above, previous studies have fabricated porous complex structure by SEBAM method in low energy density parameters. The reasons for the low energy built could be the post treatment of the porous structure easily, i.e., it is commonly seen that the support structure is connected state plate for heat conduction, lower thermal deformation in normal energy density built [12], but it is difficult to remove the support of the porous structure. Therefore, the low energy density was applied to build the porous structure without the support structure. The effect of microstructure and mechanical property were reported during SEBAM process by in changing various parameters [13,14,15,16,17,18]. Above the line energy density 0.18 J/m built, the minimum relative density of 99.5% was observed [13]. Optimized overlap distance were 0.25–0.75 mm without fused line defects and the average value of microhardness was 360 HV [14]. However, over this range, it decreased ~340 HV on average. Different built orientations in XY, ZX, ZY, XY 30° and XY 60° inclination to the start plate, the microhardness, nearly 330 HV in the ZX-P specimen, was higher than the value ~275 HV in ZY [15]. With increasing build height and thickness, the tensile strength was decreased 3% in the top [16] and the thicker one had a lower microhardness of 320 HV than 362 HV of the thin sample [17]. The impeller was built with double influence factors to effective the microhardness with more than 10% change was reported [18], the microhardness between ~340 HV to ~360 HV depended on the locations of impeller. The unique microstructure may be due to complex cyclic thermal process in three main stages as rapid cooling from the molten state to the layer temperature, followed by a near isothermal stage at the local temperature until completion of the build, and finishing with a slow cooling to the room temperature [19,20]. In summary of microstructure evolution as mentioned above, micro size β grain transformation and α′ formation in the first stage, followed by the α′ martensitic being fully or partially transformed into an α + β structure. Moreover, the previous melted layer experiences a thermal history for each subsequent layer which will result in different microstructure depends on phase transformation path during the cooling. However, in previous research showed one parameter with one working process or the many samples compared with one specific parameter. In our experiments demonstrated that 3 parameters in one sample to indicative one workpiece made by more reasons to modify the finally commercial products manufacture [13,14,15,18,21].

It is worth noting that most efforts were made to successfully develop optimized porous structures for medical metallic implants with suitable combination process parameters in order to obtain better bone osseointegration and avoid the stress shielding effect. For example, as shown in Figure 1, the geometry of orthopedic implants is divided into solid and porous parts in gradient energy density built for specified purpose by using SEBAM process. The solid structure built in high density built for dense and strength purpose, but built in low energy built for non-support demands. Transition zone between solid and porous structure, namely of overlap zone with built in mixed high energy and low energy density. Therefore, gradient energy has been widely used for porous medical metallic implant production. To the best of the authors’ knowledge, no study has been published to date on the overall characterization of gradient energy built parts, though, there are medical industrial demands for such parts fabricated by SEBAM. The specific research question this study addressed concerns the comprehension process of the effect on the microstructure and mechanical properties in gradient energy density built by SEBAM process. In this study, gradient energy density including three different of energy density was selectively built in different zones (top, middle and top) of one rectangular sample by SEBAM method using Ti6Al4V (ELI) alloy powders on the effect of the microstructure and mechanical properties. It is supposed to be of significance to evaluate the performance of SEBAM-built parts. More importantly, a thorough understanding of gradient energy on microstructure and mechanical properties obtained from this study will aid to further propel the practice application of SEBAM-built metallic parts.

## 2. Materials and Methods

### 2.1. Sample Preparation

In order to investigate the microstructure and micro-harness evolution of one SEBAMed (Kaohsiung, Taiwan) metallic part was built by SEBAM method in gradient energy density. A rectangular sample which 3 mm (thickness) × 20 mm (width) × 60 mm (height) in size was used and selectively in setting three different energy density namely high energy density (HED), low energy density (LED) and overlap energy density (OED) built on bottom (0 ≤ z ≤ 20), middle (20 ≤ z ≤ 40) and top 40 ≤ z ≤ 60 of the specimen respectively along the z axis building direction as shown in Figure 2. The energy density parameters in detail were described in the next section.

### 2.2. Selective Electron Beam Additive Manufacturing

The Extra Low Interstitial Ti-6Al-4V (Grade 23) powder supplied by Arcam AB company was used; the powder size distribution was quoted as 45 to 100 μm. The chemical composition of Ti-6Al-4V powder was supplied as showed in Table 1 followed as ASTM F3001. Recycling of non-melted powder and/or sintered powder (Figure A1 in Appendix A) was achieved via powder recovery system and a mechanical vibrating sieve which mesh size ≤ 150 μm. All SEBAM parts were fabricated on the Arcam Q10 machine (Producer: Arcam EBM GE Additive company, Gothenburg, Sweden). The SEBAM process was implemented in the vacuum chamber with pressure below 5 × 10^−3^ mbar in the beginning and finished with a pressure of 2 × 10^−5^ mbar. Each layer was setting preheated 730 °C by fast scanning with the defocused electron beam and layer thickness of 50 μm. Gradient energy density in building parameters was set as follows: the standard parameter of the beam current from 3 mA to 15 mA and scan speed from 1500 mm/s to 4530 mm/s with fixed layer thickness 50 μm, beam diameter of 100 μm and hatch spacing of 150 μm were performed. The energy density can be approximated as:(1)$$E(J/\mathrm{mm}3)=\frac{V\left(\mathrm{kV}\right)\times C\left(\mathrm{mA}\right)}{t\;\left(\mathrm{mm}\right)\times h\left(\mathrm{mm}\right)\times S\left(\mathrm{mm}/s\right)}$$
where energy density E, operation voltage V, beam current C, layer thickness t, hatch distance h and scan speed S in Table 2. The high energy density (HED) resulting from the reduced scan speed and increased beam current was therefore 63% higher compared to low energy density (LED) setups. For the transition overlapping 2 mm in width was applied overlap energy density (OED), i.e., HED in the first scan and LED in the after.

### 2.3. Microstructure Observation

Optical microscopy and scanning electron microscopy (SEM) of SEBAMed samples prepared by grinding and polishing- diamond abrasive disc with water used as coolant in the grinding process and polished with silicon carbide and 0.3 μm Al_2_O_3_ suspension. Optical and SEM samples were etched in 2.5% HNO_3_ + 5% HCl + 92.5% ethanol reagent. Secondary electron image SEM was carried on JEOL JSM-6380 (Tokyo, Japan) at 15 kV. Phase analysis was conducted using X-Ray Diffraction (XRD) (Panalytical B.V., Almelo, The Netherlands) and transmission electron microscopy (TEM) (PHILPLIES CM200 and JEOL 2000 FX, Tokyo, Japan). The characteristic CuKα radiation (λ = 1.5412 Å) in the 2θ range from 20 to 100 degrees having voltage 40 keV and current 40 mA was used. TEM samples were prepared using the following procedures: thin samples sections were manually ground to 0.06 ± 0.02 mm in thickness with silicon carbide paper and then the thin foils for TEM were electro polish using Automatic Twin-Jet Electropolisher Model 110 (Yokohama, Japan) at 25 V in a solution bath consisting of 12% perchloric acid, 15% glacial acetic acid and 75% ethanol reagent.

### 2.4. Mechanical Properties

Vicker microhardness (500 g, 15 s hold) of 15 individual measurements in each HED, OED and LED were performed on the metallographic samples using an Akashi MVK-H100 (Osaka, Japan) machine. The tensile tests on an MTS-10t (Eden Prairie, MN, USA) Tester using cylinder specimen with gauge length of 30 mm of specimen 3 (Figure A2 in Appendix B) according to ASTM-E8M specifications. The specimens were subjected to impact testing with the use of a Tinius Olsen model IT 504 polymeric impact testing machine (Tinius Olsen, Horsham, PA, USA). All tests were conducted at a pendulum capacity of 15 J, a drop height of 609.6 mm and velocity of 3.46 m/s.

## 3. Results

### 3.1. Effect of Gradient Energy Density on the Defects

Figure 3 shows representative specimens of the gradient energy from high energy density to low energy density built zones along the z axis building direction. The irregular pores which size exceeds 100 μm was only observed in the LED built specimen (Figure 3a) which is caused by the lack of fusion. Above line energy density more than 100 J/m is necessary for full densification of Ti6Al4V and below 100 J/m SEBAMed specimens contain more than 1% porosity as earlier reported [13]. This means that irregular pores did not obviously exist both in the HED and OED built zones, implying that highly dense sample can be fabricated by the SEBAM process if the energy density were more than 100 J/m. The irregular pores would greatly reduce yield stress, tensile stress and ductility [13].

In contrast the typical spherical pores several μm in size were found in all HED, OED and LED built samples in Figure 3b–d. It means no relation between the spherical pores and energy density and is mainly caused by the entrapped argon gas inside the Ti6Al4V powder during the production of plasma wire gas atomization [22]. The presence of limited small pores cannot be eliminated by hot isostatic press treatment [23] and did not significantly affect tensile properties. It might be decreased fatigue life.

### 3.2. Effect of Gradient Energy Density on the Microstructure Evolution

In order to understand the influence of the gradient energy density on three different sites, the microstructure was studied along the build direction. Figure 3b shows partially acicular α′ martensitic and α + β were developed by LED built zone where is at the top of the specimen. At the middle of the specimen built in the OED parameter, a lamella mixture of α+β structure was observed (Figure 3c). A coarser α+β lamella structure in the prior β grain was also observed at the bottom where is built in HED parameter (Figure 3d). No obvious difference in SEM analysis was found both in OED and HED built zones. However, the acicular α′ martensitic only was observed in LED built zone, which indicates cooling rate in LED zone was higher than OED and HED built zones.

Figure 4 shows the XRD analysis of the gradient energy density built by SEBAM method using Ti6Al4V powder. The characteristic peaks of dominant α phase with (101), (002) and small fraction of β phase with (110) were observed in all energy density built zones. Minor fraction of β phase in both OED and LED built zones compared to HED built zones. It is difficult to distinguish α and α′ phase by XRD analysis. Based on the mentioned above, the acicular α′ martensitic phase was only found in the LED built zone where the present characteristic peaks of α phase overlapped with α′. Interestingly, the intermetallic phase Ti_3_Al was observed in HED built zone. Further TEM analysis in Figure 5 revealed that the nano-size intermetallic Ti_3_Al dispersed near the α grain boundary, and a fine lamella α+β structure. Moreover the crystallographic relationship between α, Ti_3_Al and β phase as [001]_α_ // [001]_Ti3Al_ and [011]_α_ // [001]_β_ are similar results to what Barriobero-Vila reported of selective laser melting with intrinsic heat treatments of Ti6Al4V [24]. The analysis of the formation of nanosize Ti_3_Al without the post heat treatment or repetitive melting (intrinsic heat treatment) during the SEBAM process will be discussed in the following section.

### 3.3. Effects of Gradient Energy Density on the Mechanical Properties

A continuous microhardness on the HED, OED and LED built zones were measured in Figure 6. Apparently, it was divided into two groups via gradient energy density built. The microhardness values in average 320 to 350 HV also revealed no obvious difference between HED and OED built zones. Nevertheless, a relatively high microhardness (from 375 to 420 HV) was observed in the LED built zone. The higher microhardness in the LED built zone was caused by the acicular α′ martensitic microstructure (Figure 3b) and it is consistent with those reported for selective laser melting built Ti6Al4V alloy [25,26]. Moreover, the measure average α lath width in the HED and OED built zones were determined to be 0.69 ± 0.04 μm and 0.59 ± 0.14 μm respectively. The result is in good agreement with no obvious difference in mircohardness in the two zones. However, the microhardness in the OED built (double melting) zone showed the finer α+β lamella and grain bound of α structure than in the HED built zone. The result was in contrast to the paper showed the coarsening microstructure was caused by double melting in the same build height [14]. This implies that double melting is not only seen to be relevant to grain size. The effect of thermal history from the different building height might have a powerful impact on grain size [18].

Stress-strain curves and impact energy of three different energy density built samples were shown in Figure 7. The ultimate tensile strength (UTS) estimated 228 MPa of the LED built sample lower obviously compared to the OED built and HED built samples in Table 2. The presence of the obvious plastic deformation both in OED and HED curves were distinct with LED curve which almost no plastic deformation was observed. Both OED and HED curves had no obvious difference from the UTS and elongation, however in the finally the HED curve was slightly reduced the strain. Higher elastic modulus of OED and HED built samples compared to LED built sample. Interestingly, the highest hardness was in the LED built zone due to rapid cooling leading to acicular α′ martensitic but it had the smaller UTS and elongation compared to OED built and HED built sample. To further understand the variation of the mechanical properties, the fracture morphology was observed in Figure 7b–d. The transgranular ductile dimple tearing resulting from the coalescence of microvoids fracture surface was observed in both OED and HED built samples. Crack propagation of fine dimples at the tensile fracture indicated the extent of plastic deformation. Besides, according to the Hall-Petch strength mechanism, smaller grain size provided more grain boundaries, which can impede the movement of the dislocation, demonstrated as OED built sample has slightly higher tensile stress and elongation compared to HED built sample. In contrast, the fracture surface included cleavage, unmelted and partially melted powder were shown in LED built sample. This implied insufficient energy density that was unable to melt the deeper layer had a negative effect on tensile stress and elongation properties. This result is consistent with others reported [13,27].

The average fracture toughness properties in different energy density were shown in Table 3. Both OED and HED built samples showed the higher impact energy compared to the LED built sample which had the lower fracture toughness properties (2 to 3.2 J). These are similar results to the tensile stress in the fracture surface. Both fracture surfaces as transgranular ductile dimple fractures from the coalescence of microvoids in OED and HED built samples were observed in Figure 7b–f. However, the lower value of impact energy of fracture surface in LED built sample exhibited pores, unmelted powder and partial melt region (See Figure 7d,g). This means lower energy density may contributed to large area of pores and unmelted powder [13,27].

## 4. Discussion

### 4.1. Formation Gradient Microstructure on the Gradient Energy Density Built Zones

The gradient microstructure was built by gradient energy density from the bottom to top of the specimen. In this work, the microstructure of the particular zones in different energy density built will get the gradient microstructure (α′ + α + β in LED built zone, slightly finer α + β lamella structure in the OED built zone and α+β lamella with minor fraction of nanosize Ti_3_Al intermetallic in HED built zone were observed in Figure 3, Figure 4 and Figure 5). Moreover, all the specimens showed the microhardness values between 304 to 420 HV. The highest microhardness was contributed to the presence of α′ martensitic phase (Figure 3b) in LED built zone. However, no α′ phase was observed and only fine α+β lamella were observed both in the OED and HED built zones where the average microhardness of 340 HV was shown in Figure 6. The presence of limited small amount Ti_3_Al did not significantly affect the microhardness in the HED built zone compared to the microhardness in the OED built zone. In the present study, the thermal events were more complex due to a combination of the building height and gradient energy density.It is showed that the difference of previously report as varying microstructure only in changing building direction [15] process parameters [28,29], building high [18,30], overlap distance [14], heat treatment [19], intrinsic heat treatment [24]. Therefore, the analysis of formation of gradient microstructure/phase in gradient built zones will be discussed in the following section.

#### 4.1.1. Formation α′ Martensitic in LED Built Zone

The thermal gradient, G (K/m), and solidification velocity, R (m/s) can be used to predict solidification microstructure where G/R controls the mode of solidification (morphology) and G × R (cooling rate) in SEBAM process [31]. For an increasing energy density, the cooling rate became slower as was shown in solidification map for SEBAM Ti6Al4V melted [31]. Moreover, the cooling rate of the top was higher than the bottom also reported in SEBAMed Ti6Al4V parts [18]. In the present study, the SEM investigations in Figure 3 revealed that formation of acicular α′ martensitic in LED-built zone where at the top of the specimen along the build direction. Both the low energy built and the top of the building height caused a high cooling rate. Moreover, during the SEBAM process, as the electron beam scanned over the Ti-6Al-4V powder, a melt pool formed and then was rapidly solidified into β grains. The prior β grains underwent a rapid cooling (a critical cooling rate of >410 °C/s) transforming into martensitic α′ phase. Moreover, it its known that the build temperature of 600~650 °C in the chamber was below martensitic start temperature (Ms) 800 °C. Thus, α′ martensitic phase should be formed after melting.

#### 4.1.2. Formation α+β in OED Built and HED Built Zones

The observation of the α+β lamella structure inside OED built (Figure 3c) and HED built (Figure 3d) samples also proves the aforementioned process. Following the above mentioned formation of martensitic α′, β phase was transformed into an α′ (β → α′) reaction on cooling. In addition, α′ martensitic will decompose into α and β phases, i.e., α′ → α + β, when it is subjected to isothermal annealing in the α + β two phase field. It was reported that α′ in the thicker sample would be able to decompose into α/β phases due to higher thermal mass causing the slower cooling. During the EBM process, an electron beam was constantly scanning over the Ti6Al4V powder bed and the heat was mainly transferred from top to bottom through the built parts, the temperature of the newly deposited layers must be higher than 650 °C in the middle and bottom of the specimen that was sufficient to enable α′ decomposition.

#### 4.1.3. Formation Ti_3_Al in HED Built Zones

The formation of the intermetallic Ti3Al phase along the building direction of the bottom of the specimen was revealed using X-ray diffraction by the presence of (001) characteristic peak as shown in Figure 4 as well as by select area diffraction carried out during TEM investigations (Figure 5). According to previous investigations of SLMed Ti6Al4V and SLMed Ti6Al4V with intrinsic heat treatment, the driving force for Ti_3_Al formation due to the fast solidification during the SLM process, concentration of aluminum and oxygen [24]. Moreover, Al depletion at the interface during EBM process was reported [29]. In this work, diffraction pattern evidence of Ti_3_Al was only observed in HED built zone (the bottom of the specimen) mostly affected by heat transfer from the following layer. Since precipitation of this phase can occur during aging at 500–600 °C for several hours [31], it is reasonable to conclude that this range of temperatures was reached by the scanning of following layers, lead to the precipitation of an intermetallic Ti_3_Al.

### 4.2. Cooling Path in Gradient Energy Density and Subsequence Heat Transfer

From the microstructure point of view as mentioned above in three different zones. The α′ + α + β in the LED built zone where it is the top of the specimen, α + β lamella structure in OED built zone where it is the middle of the specimen and α + β lamella with small amount of Ti3Al in HED built zone where it is the bottom of the specimen, were observed due to different thermal history. It implied the cooling rate of the top (LED built zone) was faster than the middle (OED built zone) and the bottom (HED built zone) of the specimen. Further the evidence of slight smaller width of α lath in the middle in Figure 3 implied slight higher cooling rate in the middle compared to the bottom. Based on the literature reported the complex thermal events in the EBM process can be simplified in three main stages [19]. The first step is a rapid cooling from the molten state to the layer temperature, followed by a quasi-isothermal stage at the local cooling until completion of the build, and finishing with a slow cooling to room temperature. Gradient energy density and different locations causing different thermal events were predicted in Figure 8 base on the microstructure analysis as mentioned above. Moreover, the heat coming from the following layer melting will transfer to the previous layer causing the similar effect as aging heat treatment was observed only at the bottom. Those observed results proved the effect combined gradient energy density and different locations on microstructure and phase transformation due to different thermal history.

## 5. Conclusions

In this work, gradient energy density was carried out on a specimen from the bottom to the top by SEBAM technology. The effect of energy density on the microstructure and mechanical properties of SEBAMed Ti6Al4V was investigated by SEM, X-ray, TEM, microhardness, tensile and impact tests. Based on the experimental results, the following conclusions can be drawn:

Gradient energy density (16 to 26.5 J/mm^3^) built sample with dimension of 20 mm × 60 mm × 3 mm was fabricated by SEBAM. Small spherical pores coming from powder were found to be independent of the energy density and locations in the built part. Large irregular pores were observed in LED built zone due to lack of fusion but no internal crack and large pores were observed in OED or HED built zones.The α′ + α + β, α + β and α + β + Ti_3_Al graded microstructure was observed from the top to the bottom. The graded microstructure was caused by the complex heat transfer due to gradient energy density and locations during building. Correspondingly, gradually increased microhardness was achieved from the bottom to the top and there was no obvious difference from the bottom to the middle.Lack of fusion defects resulted in a decrease in plasticity in the LED built zone. The unmelted powder and voids in following fracture morphology were observed by SEM.A small amount intermetallic Ti3Al was observed only in the bottom (HED built zone). The formation of Ti3Al is attributed to the heat transfer from the following layer melting.

## Figures and Tables

**Figure 1 materials-13-01509-f001:**
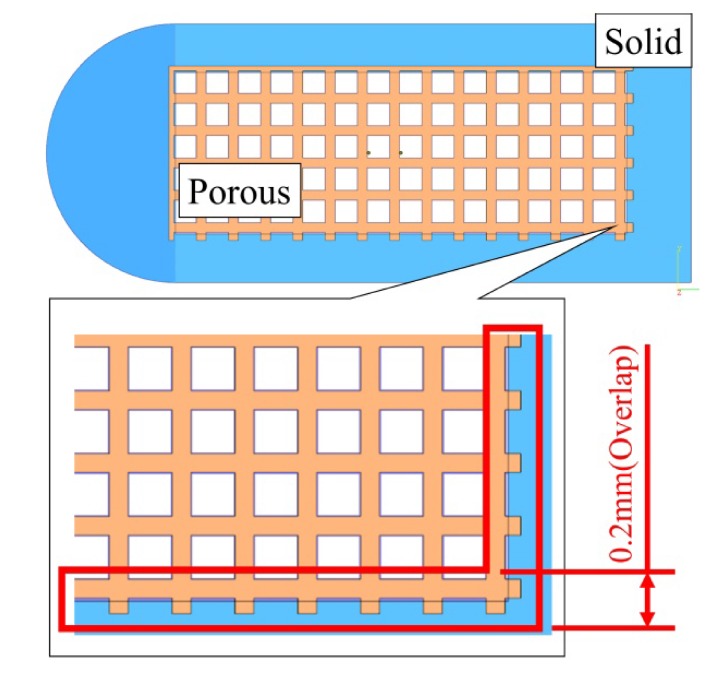
Schematic diagram of commercial porous fusion cage structure was fabricated by selective electron beam additive manufacture (SEBAM) method using gradient energy density on three different zones. Solid and porous parts were made independently by higher and lower energy density. The overlap site with 0.2 mm in width between solid and porous parts was made sequentially by higher and lower energy density.

**Figure 2 materials-13-01509-f002:**
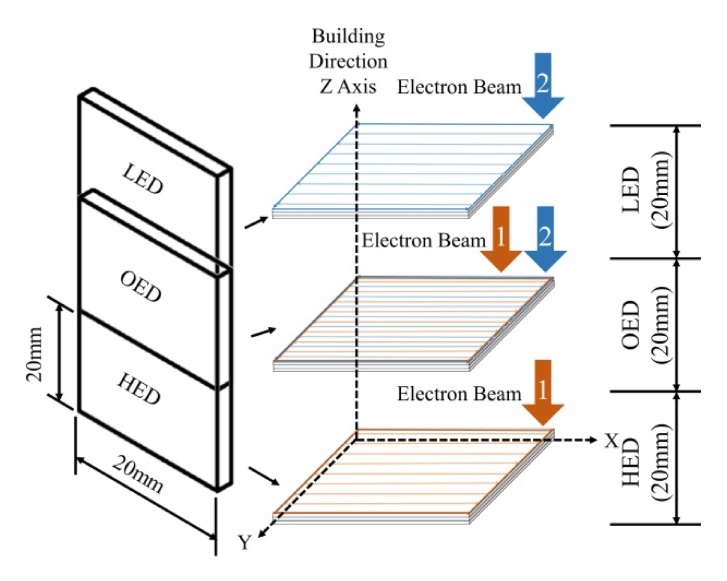
Schematic of SEBAMed samples made by higher, lower and overlap energy density from the bottom to top which labeled as high energy density (HED), overlap energy density (OED) and low energy density (LED) along build direction fabricated by electron beam additive manufacture. HED, LED built zones independently 26.5 and 16 J/mm^3^. OED built zone in 26.5 J/mm^3^ first and build sequentially 16 J/mm^3^.

**Figure 3 materials-13-01509-f003:**
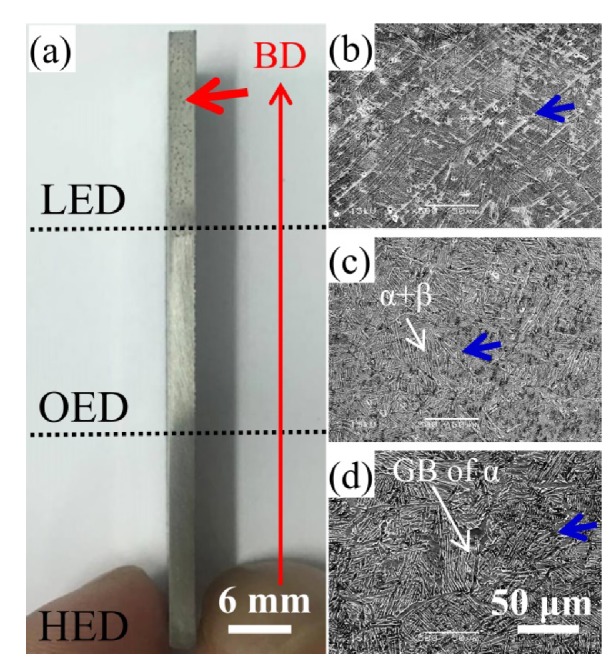
Side view of specimen from bottom to top along the build direction (BD) denoted in red arrow of (**a**) showing the high visible pores which range from several hundred μm to several mm in size denoted by red arrow in LED built zone. SEM secondary electron images shows smaller pores several μm in size denoted by blue arrows in (**b**) LED, (**c**) OED and (**d**) HED built zones. Besides, α+β lamella and grain bound (GB) of α structure denoted by white arrows both were observed in (**c**) OED and (**d**) HED built zones.

**Figure 4 materials-13-01509-f004:**
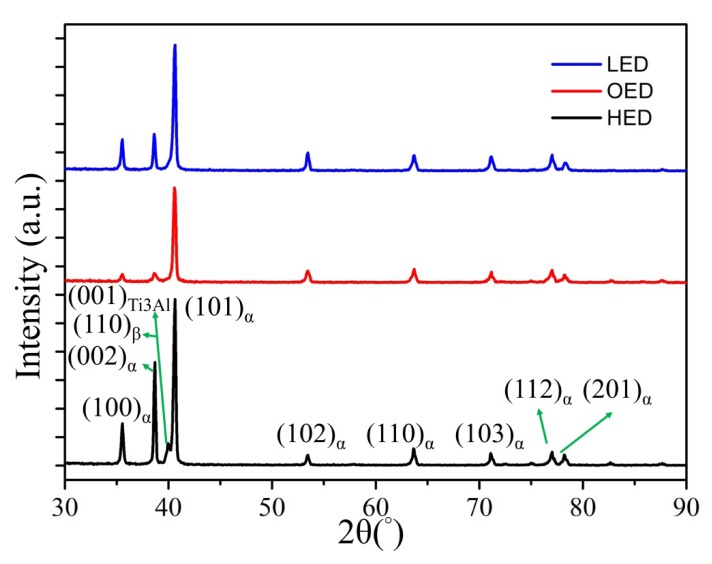
The X-ray diffraction spectrum of HED, LED and OED built samples. The α-Ti6Al4V characteristic (100), (002) and (101) and β-Ti6Al4V characteristic (110) peaks were observed in all samples. The characteristic peaks of α phase are overlapped with α′ martensitic phase only in LED built sample. Minor Ti_3_Al phase of its characteristic peak (110) was observed only in HED built sample.

**Figure 5 materials-13-01509-f005:**
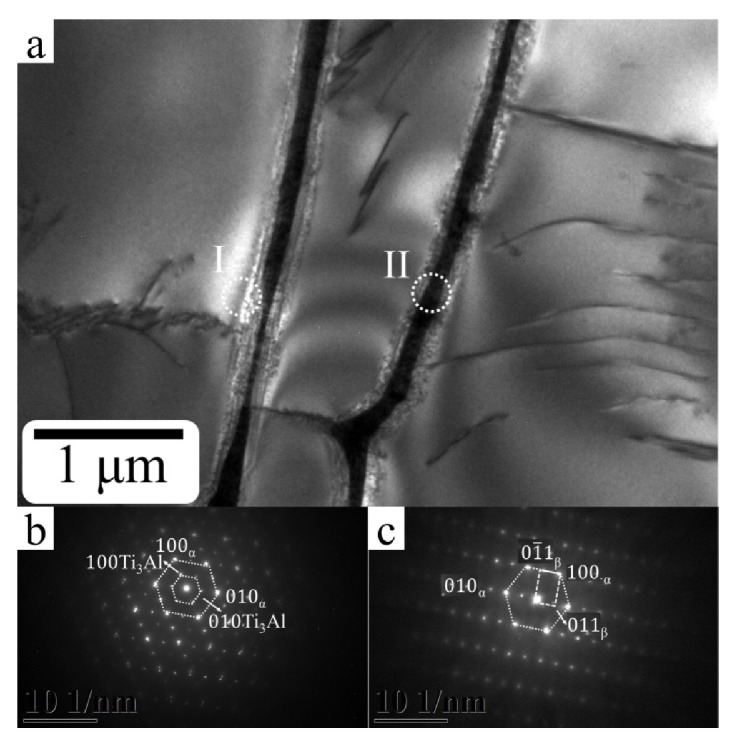
(**a**) Bright field image of ultrafine α+β lamella microstructure found at the center of HED sample. (**b**) Select area diffraction patter from the cycle region (I) in (**a**) showing the present Ti3Al within an α lamella matrix with crystallographic relationship [001]_α_//[001]_Ti3Al_ (**c**) Select area diffraction patter from the cycle region (II) in (**a**) showing the ultrafine α + β lamella microstructure with crystallographic relationship [001>]_α_ // [011]_β_.

**Figure 6 materials-13-01509-f006:**
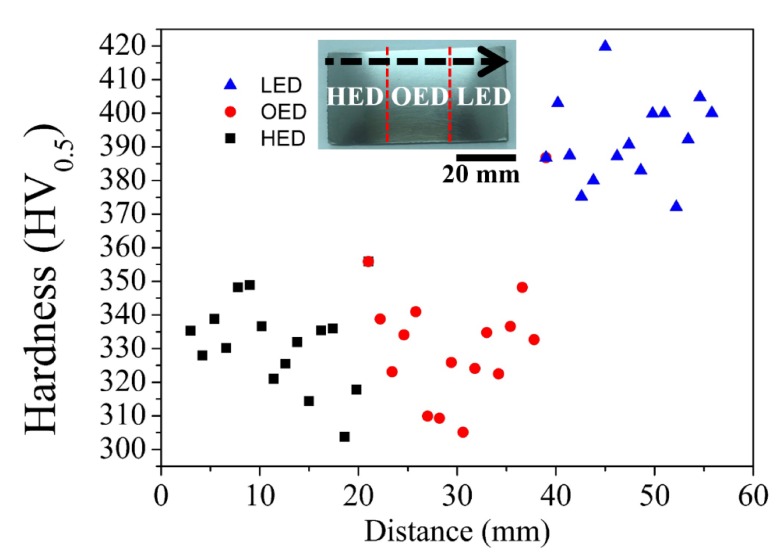
Vickers microhardness profile was measured from bottom to top (black arrow) of the cross section specimen which were built by gradient energy. HED, OED and LED built zones are indicated by black, red and blue symbols.

**Figure 7 materials-13-01509-f007:**
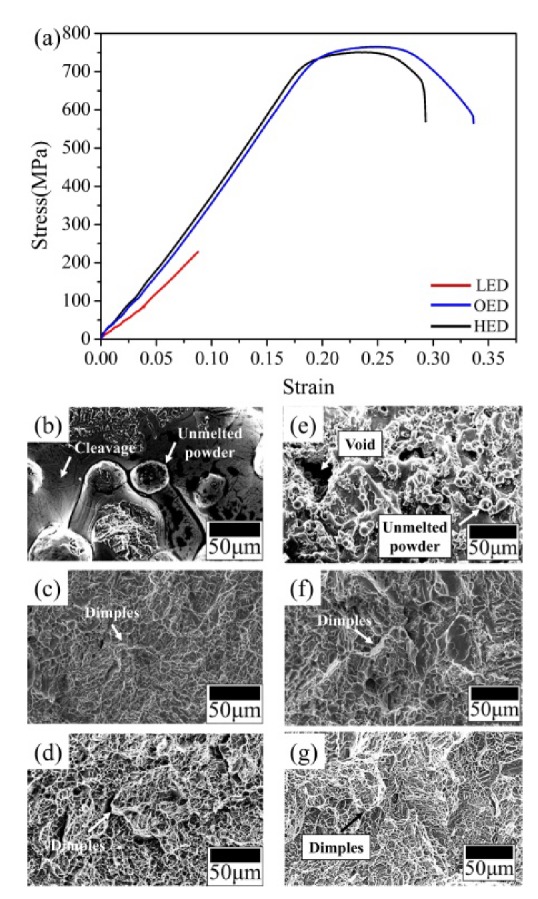
(**a**) Tensile stress-strain curves of gradient energy built samples. LED, OED and HED built samples were respectively denoted by red, blue and black curves. Representative fracture surface of tensile test of different density built in (**b**–**d**). (**b**) Unmelted powder and brittle cleavage in LED built sample, (**c**,**d**) both tensile fracture morphologies show fine and deeper dimples in HED and OED built samples. Representative fracture surface of impact test of different density built in (**e**–**g**). (**e**) Unmelted powders and voids were observed in LED samples, (**f**,**g**) both impact fracture morphologies show fine and deeper dimples in HED and OED built samples.

**Figure 8 materials-13-01509-f008:**
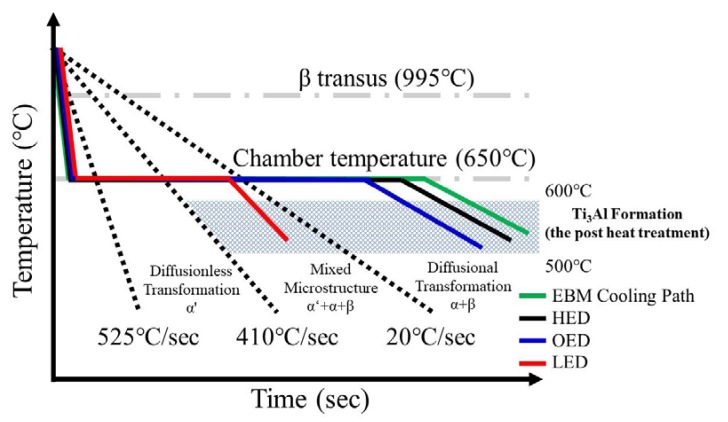
Prediction cooling path and phase transformation based on the literature [19] of LED, OED and LED parameters built in top, middle and bottom of the specimen respectively. The difference in cooling rate result from different energy density and spatial location thus contribute to different microstructure. i.e., α′ + α + β in LED built zone, α + β lamella structure in the OED built zone and α + β + Ti_3_Al intermetallic phase in the HED built zone were observed.

**Table 1 materials-13-01509-t001:** The chemical composition (in wt %) of Ti-6Al-4V powder.

Element	Ti	Al	V	C	Fe	O	N	H	Y
ASTM F3001	Bal.	5.50–6.50	3.50–4.50	≤0.08	≤0.25	≤0.13	≤0.05	≤0.012	≤0.005
Powder	Bal.	6.49	4.01	0.02	0.18	0.07	0.02	0.001	<0.001

**Table 2 materials-13-01509-t002:** Main process parameters with fixed layer thickness 50 μm, beam diameter of 100 μm and hatch spacing of 150 μm used for the SEBAM process.

Items	Scan Speed	Beam Current	Volume Energy Density
HED	(Scan1) 4530 mm/s	15 mA	26.5 J/mm^3^
LED	(Scan2) 1500 mm/s	3 mA	16.0 J/mm^3^
OED	(Scan1) 4530 mm/s (Scan2) 1500 mm/s	15 mA 3 mA	26.5 J/mm^3^ 16.0 J/mm^3^

**Table 3 materials-13-01509-t003:** Mechanical properties of SEBAMed Ti6Al4V in varying energy density.

Sample	UTS (MPa)	YS (MPa)	Young’s Modulus (GPa)	Elongation (%)	Impact (J)
LED	228 ± 10	NA	74 ± 3	8 ± 1	2.7 ± 0
OED	765 ± 11	725 ± 10	108 ± 1	31 ± 1	12 ± 0
HED	752 ± 19	737 ± 3	116 ± 8	26 ± 1	13 ± 1

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
