# Peer review of "Effect of Gradient Energy Density on the Microstructure and Mechanical Properties of Ti6Al4V Fabricated by Selective Electron Beam Additive Manufacture"

_materials, 2020, doi:10.3390/ma13071509_

Round 1

Reviewer 1 Report

Overall the paper is of interest to the journal readership and contains novel findings. There are a few minor suggested changes that I believe should be address before publication:

  • Introduction: more specific information should be provided on what manufacturing parameters have previously been utilised for EBM in the literature and further clarity on relationships between key properties
  • Language should be checked throughout the manuscript - in a number of areas the authors could be more precise
  • Methods section is very clear
  • Overall the results are clearly presented but a few suggestions:
    • Figure 6: why have you joined these data points? The data is not continuous, perhaps it is more relevant to bit trend lines or leave as a scatter?
    • Table 3: are these average values or single measurements? If only single samples were tested then this probably needs repeating to have a minimum of n=3

Author Response

Overall the paper is of interest to the journal readership and contains novel findings. There are a few minor suggested changes that I believe should be address before publication:

1.Introduction: more specific information should be provided on what manufacturing parameters have previously been utilised for EBM in the literature and further clarity on relationships between key properties

 A: Thanks for your suggestion. The related literatures regarding the               mechanical properties and various parameters was provided in line 58-67.

2. Language should be checked throughout the manuscript - in a number of areas the authors could be more precise

A: Thanks for your correction. We have checked the language and rechecked by the person who use English as native language.

3.Methods section is very clear.

A: Thanks for your comment.

Overall the results are clearly presented but a few suggestions:

4.Figure 6: why have you joined these data points? The data is not continuous, perhaps it is more relevant to bit trend lines or leave as a scatter?

A: Thanks for your comment. The sample was combined HED, OED and LED built zones and was measured microhardness from the HED to LED built zones. The figure 6 with scatter points was updated in page 8 and the figure legend was updated in line 243 to 245. 

5. Table 3: are these average values or single measurements? If only single samples were tested then this probably needs repeating to have a minimum of n=3

A: Thanks for your comment. All the data was test at least 3 times and the standard deviation was add in table 3 in line 275 in page 9.

Reviewer 2 Report

The article is interesting and the results presented in it are interesting, but after reading it, there are questions that the authors should answer.

Was the powder used in the research new or used?

Have you checked the powder morphology?

Table 3 - please add deviation for the results in this table.

What was the size and shape of samples for mechanical testing? 

Was the shape of the samples for mechanical tests printed or machined?

Was the surface of the samples post-processed or was it as-built?

What is the porosity of LED, OED, HED in %?

What was the surface roughness of the samples?

Author Response

Comments and Suggestions for Authors

The article is interesting and the results presented in it are interesting, but after reading it, there are questions that the authors should answer.

1.Was the powder used in the research new or used?

A: Thanks for your question. The used powder was conduct in this study. Follow our internal information, the oxygen content of the fresh powder was 700 ppm. After 10 times recycle, the oxygen was increased 900 ppm. It is still lower than 1300 ppm in the ASTM F3001 standard specification for additive manufacturing Titanium-6 Aluminum-4 Vanadium ELI (extra low interstitial) with powder bed fusion. The composition of the powder was shown in table 1.

2.Have you checked the powder morphology?

A: Thanks for your question. The used powder was shown in spherical shape in appendix 1 in line 396 to 399 in page 13. Non spherical powder decreased flowability has negative effect on powder bed fusion. 

3.Table 3 - please add deviation for the results in this table.

A: Thanks for your comment. All the data was test at least 3 times and the standard deviation was add in table 3 in line 275 in page 9.

4.What was the size and shape of samples for mechanical testing? 

A: Thanks for your comment. The tested sample was cylinder shape with 30 mm in gauge length, 6.0 mm in diameter, 6.0 mm in radius and 36 mm in length of reduced parallel section. The size and shape of the tensile test was corrected in line 156 and 157.

5.Was the shape of the samples for mechanical tests printed or machined?

A: The machined respectively LED built sample was shown in appendix 2 in line 400 to 403 in page 14.

6.Was the surface of the samples post-processed or was it as-built?

A: All the data in this study was conducted as built samples without any post treatment. Only the tensile test samples with CNC machining for the final size for tensile test.

7.What is the porosity of LED, OED, HED in %?

A: The porosity measurement was conducted by Archimedes method. 0.08 % in HED, 0.03 in OED and 1.80 % in LED.

8.What was the surface roughness of the samples?

A: The surface roughness are slight difference in three types of sample, 9.59±0.35 μm in HED, 10.02±0.25 μm in OED and 10.56±0.24 μm in LED.

Round 2

Reviewer 2 Report

Autors answered all questions.